# Inositol Pyrophosphate Pathways and Mechanisms: What Can We Learn from Plants?

**DOI:** 10.3390/molecules25122789

**Published:** 2020-06-17

**Authors:** Caitlin Cridland, Glenda Gillaspy

**Affiliations:** Department of Biochemistry, Virginia Tech, Blacksburg, VA 24061, USA; cridc2@vt.edu

**Keywords:** inositol phosphate, inositol pyrophosphate, inositol, inositol phosphate signaling, PPIP5K, ITPK

## Abstract

The ability of an organism to maintain homeostasis in changing conditions is crucial for growth and survival. Eukaryotes have developed complex signaling pathways to adapt to a readily changing environment, including the inositol phosphate (InsP) signaling pathway. In plants and humans the pyrophosphorylated inositol molecules, inositol pyrophosphates (PP-InsPs), have been implicated in phosphate and energy sensing. PP-InsPs are synthesized from the phosphorylation of InsP_6_, the most abundant InsP. The plant PP-InsP synthesis pathway is similar but distinct from that of the human, which may reflect differences in how molecules such as Ins(1,4,5)P_3_ and InsP_6_ function in plants vs. animals. In addition, PP-InsPs can potentially interact with several major signaling proteins in plants, suggesting PP-InsPs play unique signaling roles via binding to protein partners. In this review, we will compare the biosynthesis and role of PP-InsPs in animals and plants, focusing on three central themes: InsP_6_ synthesis pathways, synthesis and regulation of the PP-InsPs, and function of a specific protein domain called the Syg1, Pho1, Xpr1 (SPX ) domain in binding PP-InsPs and regulating inorganic phosphate (P*i*) sensing. This review will provide novel insights into the biosynthetic pathway and bioactivity of these key signaling molecules in plant and human systems.

## 1. Introduction

Inositol phosphates (InsPs) are a widely known family of signaling molecules, with conserved functions in numerous cellular processes in eukaryotes, many of which are related to human health [1,2,3]. The InsP family of molecules consists of monophosphorylated inositol (InsP) to the fully phosphorylated InsP_6_ (inositol hexakisphosphate), and are generated by InsP kinases that reversibly add phosphates to specific positions of the 6-carbon inositol ring backbone [2,4]. The specific number and position of phosphates on the inositol ring is thought to confer specific information within the cell, thus InsPs are thought to function as a type of chemical signaling language first described in [5], and elaborated on in plants [6]. In addition, InsPs can be further phosphorylated to produce the high-energy inositol pyrophosphates (PP-InsPs), which while only found in low concentrations in the cell, have been implicated in an array of developmental, metabolic and signaling processes [7]. Given the emerging role of PP-InsPs in controlling eukaryotic signaling pathways, it is important to consider how disrupting Ins and InsP biosynthesis for therapeutic uses impacts PP-InsPs and the processes they mediate.

One significant difference between plants and animals is the large phosphate (P) storage capacity of InsP_6_ present in seeds. In the model plant *Arabidopsis thaliana*, for example, InsP_6_ comprises ~1% of seed dry weight [8]. This large pool of stored P is important for maintaining metabolism between the phases of seed germination and for the onset of efficient photosynthesis [9]. A second potentially critical difference is the presence of a known InsP_6_ transporter in plant cells called Multidrug Resistance Protein 5 (MRP5) [10]. The MRP5 gene product of *Arabidopsis* encodes an ABC transporter that most likely functions to transport InsP_6_ across the vacuolar membrane during seed development, allowing high amounts of InsP_6_ to be stored in the seed [10]. At present we have a good understanding of the synthesis and transport of InsP_6_ within plants, due to the presence of key *Arabidopsis* mutants that are defective in InsP kinases [11] or the *Atmrp5* gene [10]. A third major difference is that plants have no known Ins(1,4,5)P_3_ receptor that could serve to release intracellular calcium stores in response to signaling-stimulated hydrolysis of phosphatidylinositol(4,5)P_2_ (PtdIns(4,5)P_2_) [12]. These key differences highlight the fact that plants and animals likely had different pressures driving evolution of central genes in the InsP pathway. The purpose of this review is to compare knowledge on the InsP and inositol pyrophosphate (PP-InsP) pathways in humans and plants, and highlight how new information on the plant InsP pathway may provide insights into the function of this pathway in other eukaryotes. In this review, we will concentrate on three central themes for our comparison of plants and humans: InsP_6_ synthesis pathways, synthesis and regulation of the PP-InsPs, and function of a specific protein domain called the SPX domain in binding PP-InsPs and regulating inorganic phosphate (P*i*) sensing.

## 2. Myo-Inositol: The Signaling Scaffold and Precursor for the Lipid Independent Pathway

Similar to other eukaryotes, the scaffold for InsPs is the most abundant inositol isomer in plants, the so-called myo-inositol isomer. Herein, we will refer to myo-inositol as “inositol”. Plants also contain D-chiro, epi-, neo-, muco-, and allo-inositol isomers in minor amounts [13], and these may be important for various physiological events. For example, the presence of D-chiro-inositol has been linked to salt stress tolerance in plants [14,15]. Intriguingly, a lack of D-chiro-inositol has been correlated with various diseases in humans, such as polycystic ovary syndrome and type 2 diabetes [16,17,18,19,20,21]. It is also interesting to note that groups have proposed supplementing patient diets with D-chiro-inositol rich foods and supplements as a form of treatment [22].

## 3. Is the Lipid-Independent InsP_6_ Synthetic Pathway Plant Specific?

A canonical InsP_6_ synthesis pathway is shared between plants and animals, and is usually referred to as the “Lipid-Dependent” pathway, as the first step is initiated by the hydrolysis of PtdIns(4,5)P_2_ into Ins(1,4,5)P_3_ and diacylglycerol (DAG) [23,24,25] (Figure 1). This Ins(1,4,5)P_3_ product can be sequentially phosphorylated by two sets of enzymes in both plants and animals, namely the Inositol polyphosphate Multikinases (IPMK; EC 2.7.1.151), also known as IPK2 in plants, and the Inositol Pentakisphosphate 2-Kinase (IPK1; EC 2.7.1.158) [23,24,26].

InsP_6_ can also be synthesized from the phosphorylation of inositol by a series of InsP kinases in a pathway previously thought to be unique to plants and slime molds: the “Lipid Independent” pathway of InsP_6_ synthesis (Figure 1). In this pathway, inositol is first acted on by an enzyme named *myo*-Inositol Kinase (MIK; EC 2.7.1.64). The *Arabidopsis* genome encodes one copy of the Mik gene, and the MIK enzyme is known to phosphorylate the 3-position of inositol [27,28]. Atmik mutants have a significant reduction in InsP_6_ seed levels, indicating that a portion of seed InsP_6_ requires the function of MIK. However, InsP_6_ levels in vegetative tissues from Atmik mutants have not yet been reported [27]. In maize (corn), mik knockout mutants accumulate 50% less seed InsP_6_, and contain a reduction in other InsPs [28], so the function of MIK in seed InsP_6_ synthesis is thought to be conserved across various plant species. The human genome, in contrast, does not contain a highly homologous MIK orthologue although one human ribokinase and several adenokinases contain limited regions of homology to MIK. We note that animals may not require MIK for a lipid -independent pathway as the *myo*-inositol phosphate synthase (MIPS; EC 5.5.1.4) can convert Glucose-6-P to Inositol 3-P [29], bypassing the need for MIK (Figure 1). An alternative is that there is a human MIK that is distinctly different compared to the plant MIK. 

In plants, the enzyme thought to be responsible for phosphorylating InsP to InsP_2_, which was first isolated as a low phytic acid (lpa) mutant in rice and named LPA1, also lacks a clear human orthologue. Exciting new data indicate that both plants and animals have a way to directly phosphorylate Ins 3-P, without using an LPA1 enzyme [30]. Specifically, the Inositol 1,3,4-Trisphosphate 5/6-Kinases (ITPKs; EC 2.7.1.159), are enzymes that are well-conserved in plants and humans and have been examined previously for their ability to convert InsP_3_ isomers to InsP_4_ and InsP_5_ [31,32]. New data from Archaea suggested that other eukaryotes might be able to synthesize InsPs via a Lipid-Independent pathway using ITPK1 as a highly promiscuous InsP kinase [30]. In this study the authors identified 4 genes homologous to ITPK1 in the Archaea *Lokiarchaeota candidatus*, sharing an ATP-grasp domain and conserved regions in the InsP- and ATP-binding sites, yet containing a more restricted InsP binding site [30]. Using both enzyme assays and yeast complementation assays, the authors showed HsITPK1 and AtITPK1 can phosphorylate Ins(1)P and Ins(3)P to InsP_5_ in vitro and in vivo [30]. Plants also contain four ITPK genes, which exist as part of a multigene family. It is interesting to note that only two of the four *Arabidopsis itpk* loss-of-function mutants have reduced seed InsP_6_ levels, suggesting redundancy of the plant genes [27]. 

The last step in the lipid-independent pathway, conversion of Ins(1,3,4,5,6)P_5_ to InsP_6_ is catalyzed by another very well-conserved enzyme in eukaryotes, named Inositol Pentakisphosphate 2-Kinase (IPK1; EC 2.7.1.158) in both plants and animals [33,34]. Atipk1 loss-of-function mutants have an 83% reduction in InsP_6_ levels in seeds, in addition to a reduction in InsP_7_ and InsP_8_ [24,35]. The human IPK1, named HsIPPK, can be knocked down, and these mutants accumulate InsP_4_ at the expense of InsP_6_ levels [34], which is a notable difference from the situation in Atipk1 mutants.

## 4. Catalytic Flexibility in Plant and Human ITPKs 

We have already described how the plant and human ITPKs are capable of accepting multiple InsP substrates. This class of enzymes also has another unexpected role in plants. In non-plant eukaryotes, InsP_6_ kinases (inositol-hexakisphosphate kinase; IP6K; EC 2.7.4.21) catalyze the synthesis of InsP_6_ and InsP_7_ from InsP_5_ and InsP_6_, respectively [36]. However, orthologous genes encoding IP6Ks are notably absent in investigated plant genomes [37]. When searching for the enzymes responsible for the initial step in PP-InsP synthesis, conversion of InsP_6_ to InsP_7_, our group and others focused on the ITPKs, which are able to catalyze the sequential conversion of InsP_3_ to InsP_4_ and InsP_5_. The *Arabidopsis* genome contains four ITPK genes (Itpk1, Itpk2, Itpk3, and Itpk4), which encode ATP-grasp domain-containing proteins [31,38]. In humans, a sole HsItpk gene encodes an ATP-grasp containing ITPK enzyme, which shares 30% sequence similarity to its *Arabidopsis* counterpart [31,39]. HsITPK1 is responsible for the production of Ins(1,3,4,6)P_4_ and Ins(1,3,4,5)P_4_ in a ratio of 3:1, whereas AtITPKs produce the same products in the 1:3 ratio, hinting at a functional divergence in the InsP synthetic pathways in animals and plants [31,38]. Furthermore, both HsITPK1 and AtITPKs can phosphorylate InsP_4_ at the 1-position to produce InsP_5_.

Our biochemical analysis has demonstrated that both AtITPK1 and AtITPK2 are able to phosphorylate InsP_6_ in vitro to produce a more phosphorylated product [40]. This novel activity of AtITPKs is also reported in other recent studies, identifying the product of InsP_6_ phosphorylation by AtITPK1/2 as 5PP-InsP_5_ [41]. The high catalytic flexibility of plant ITPKs has likely evolved due to the paucity of canonical InsP_3_ 3-kinases (IP3Ks) [42] and IP6Ks. In animals, dedicated IP3K and IP6K families are responsible for the phosphorylation of InsP_3_ and InsP_6_, respectively [43]. Given the notable lack of IP6Ks and the closely related IP3K in plants, it is conceivable to hypothesize that AtITPKs have evolved to expand their substrate specificity to accommodate several diverse substrates. We note here that the kinase domains of the AtITPKs share structural similarity to IP3Ks and to the diphosphoinositol-pentakisphosphate kinases (PPIP5Ks; EC 2.7.4.21), despite limited sequence identity [39,44].

## 5. The PPIP5K/VIP/VIH Dual Domains are Arbiters of Biology

One focus of our lab is on the highly conserved PPIP5Ks, which are also known as VIP or VIH enzymes in plants. Until recently it was suspected that the PPIP5Ks in plants functioned in two steps to convert InsP_6_ to the first inositol pyrophosphate, called 5PP-InsP_5_ or InsP_7_, and also to convert InsP_7_ to 1,5PP_2_-InsP_4_ or InsP_8_. As mentioned before, recent work has shown the likelihood that this is not true, and that plant ITPKs are responsible for conversion of InsP_6_ to InsP_7_ [40]. This leaves the plant PPIP5K enzymes as functioning in a very similar role as the animal PPIP5Ks. Indeed both the human PPIP5Ks and the plant PPIP5Ks contain a conserved structure: an N-terminal ATP-grasp kinase domain (KD) and a C-terminal phosphatase domain (PD) [45]. Like humans, *Arabidopsis* contains two Ppip5k genes orthologous to the mammalian Ppip5ks, and both AtPpip5k genes encode catalytically active enzymes, sharing a conserved Asp residue in the KD, which is necessary for kinase activity [37,45,46]. In addition, AtPPIP5Ks and HsPPIP5Ks both share a conserved Pleckstrin-homology (PH) domain, which interrupts the C-terminal PD [47]. PH domains are found in signaling proteins and bind phospholipids and molecules derived from phospholipid head groups [48]. The human PPIP5K1 PH domain is known to bind PtdIns(3,4,5)P_3_ preferentially, and can also bind InsP_6_ and PtdInsP_2_ [49]. PH domain-containing proteins are known to interact with InsP_7_, with the inositol pyrophosphate competing with PtdInsP_2_ and PtdInsP_3_ molecules for binding [47,50,51]. A conserved arginine residue (Arg417 in HsPPIP5K1) is required for ligand binding in the PH domain in human PPIP5Ks [47]. This arginine residue is not conserved in *Arabidopsis* PPIP5Ks, however, the substituted lysine residue carries a similar charge [47]. This could indicate that the AtPPIP5K PH domain has a similar phospholipid-binding affinity, however, analysis on ligand binding has not yet been assessed in plants. In addition to sharing a conserved dual domain structure, the AtPPIP5Ks and HsPPIP5Ks also share a C-terminal intrinsically disordered region (IDR) [52,53]. IDRs are involved in protein-protein interactions, and the HsPPIP5K IDRs have been implicated in protein scaffolding and interacting with proteins involved in lipid metabolism, vesicle-mediated transport and actin cytoskeleton organization [52,53]. While a number of proteins interacting with the disordered regions in the PPIP5K family have been identified, the function of these protein-protein interactions and their involvement in PP-InsP metabolism remains unknown.

The presence of two competing domains (i.e. the KD and PD) is a unique feature of the PPIP5K enzymes, which has been discussed in detail [52]. In metabolism this is usually referred to as a futile cycle, in that the substrate, InsP_7_, for one domain (the KD), is also the product of the other domain (the PD). This dual domain structure is found in all eukaryotic PPIP5K enzymes except for those from the ancestral slime mold, *Dictyostelium discoideum* [52], which lacks a PD, either within the protein or as an independent gene, explaining the slime mold’s ability to synthesize unusually high levels of InsP_7_ and InsP_8_ [54]. While the protein structures of the HsPPIP5Ks, AtPPIP5Ks or their orthologues have not yet been elucidated, we do know that the PD of HsPPIP5Ks and the yeast orthologue, Asp1, have 1-phosphatase activity, acting on the 1PP in an InsP_7_ or an InsP_8_ substrate [55,56,57,58]. Therefore, it seems likely that the function of the PPIP5K enzymes is to interconvert InsP_7_ and InsP_8_ signaling molecules, and thus control specific information within the cell. Gu et al., performed a detailed analysis of the kinetics of the human PPIP5Ks [56]. To discern the balance between activities of the KD and PD, these authors mutated each domain within the full-length enzyme, and tested activity of each with respect to physiologically relevant substrate concentrations. Their results indicated that the PD can restrict the maximal activity of the KD under certain conditions. Interestingly, a human mutation in the PD associated with elevated PPIP5K kinase activity and autosomal recessive nonsyndromic hearing loss indicates that the KD-PD balance is required in certain instances [56]. Gu et al. also found that inorganic phosphate (Pi) is a strong inhibitor of the PD of the HsPPIP5K2 isoform [56]. This last part is of enormous importance as PP-InsPs are known to serve as the proxy for Pi sensing in different eukaryotes, including plants [59,60].

Recent work illustrates that the balance between the KD and PD is crucial for controlling growth, development and Pi sensing in plants. Zhu et al. (2019) constructed transgenic *Arabidopsis* plants carrying a copy of the full-length PPIPK (VIP1/VIH2) protein harboring a known mutation either in the KD or PD that eliminates activity of that domain [61]. The authors compared these plants to those overexpressing the intact PPIPK (VIP1/VIH2) protein–(active KD and PD), finding that the intact PPIPK plants did not show greatly perturbed growth, development or Pi sensing, as measured by the accumulation of Pi within the plant [61]. In contrast, transgenic plants overexpressing PPIPK (VIP1/VIH2) with a mutated KD are compromised in plant growth and have greatly increased Pi accumulation, consistent with an elevated phosphate starvation response or PSR [61]. Transgenic plants overexpressing PPIPK (VIP1/VIH2) with a mutated PD also have less overall plant growth, accompanied by less Pi accumulation in tissues, suggesting these plants have a repressed PSR. Thus, an unbalanced PD activity appears to turn off Pi sensing, and an unbalanced KD activity appears to turn up Pi sensing. These data strongly support the idea that plants require InsP_8_ for sensing environmental Pi, and the PPIP5Ks function to increase or decrease InsP_8_ levels in response to fluctuating environmental Pi. Given this, it is imperative to understand whether or not Pi can also regulate the plant PPIP5K PD, as is the case for the human PPIP5Ks [56]. We note that the plant PPIP5K PD has been recalcitrant to enzyme studies, although the KD has been nicely characterized [40,41]. Notably, the single report of PD activity from plant PPIP5Ks required incubation for 16 hours to visualize even very modest activity, thus more work is needed to nail down the substrate specificity of this important domain [61].

Another powerful approach in plants, the use of genetic mutants, underscores the importance of InsP_8_ in Pi sensing. *Arabidopsis* mutants devoid of AtPPIP5K activity display significant growth phenotypes and defects in Pi sensing [61,62]. These mutants have increased levels of InsP_7_ and a reduction in InsP_8_ levels, and exhibit severely stunted growth with shorter roots and smaller leaves [61,62]. In addition, these mutants exhibit Pi-stress related phenotypes, including increased Pi accumulation and upregulation of PSR genes even under sufficient Pi growth conditions [62], Additionally, other mutants defective in InsP_6_ synthesis, itpk1, ipk1 and itpk4 mutants, exhibit an upregulation of the PSR under Pi-replete conditions [35,63]. Together, these data support a critical role for InsP_8_ in the regulation of Pi homeostasis in plants, a role which may be shared in humans. In humans, PP-InsPs are known to regulate certain transmembrane Pi-transport proteins, with ip6k1 ip6k2 double mutant HCT116 cells having a decreased rate of cellular Pi-efflux, strongly suggesting that PP-InsPs are required for Pi efflux [64]. 

## 6. Do Human Signaling Pathways Share a Common Mechanism for Regulation by InsP_8_?

Up to this point we have discussed new and critical data from plants regarding regulation of Pi sensing and the PSR by InsP_8_. It is important to describe the mechanism by which PP-InsPs are thought to influence signaling pathways, and to ask whether this mechanism is shared by humans.

PP-InsPs have been likened to a “molecular glue”, acting to facilitate protein-protein interactions and initiate energy- and nutrient-mediated responses. One protein domain found throughout eukaryotes that PP-InsPs bind to is named the SPX domain, named after the yeast proteins, Syg1 and Pho1, and the mammalian Xpr1. SPX-domains contain a basic surface of lysine residues that are known to coordinate the binding of PP-InsPs [65,66]. For many years it was known that the SPX domain basic surface could bind Pi. This, along with the identification of yeast and plant mutants defective in a variety of SPX domain-containing proteins, prompted the hypothesis that the SPX domain, itself, was the receptor for Pi sensing [67,68,69,70]. Wild et al. disrupted the thinking on this issue in 2016 by discovering that PP-InsPs can also serve as ligands for SPX domains, binding with high affinity (50 nM-100 μM) [65], thus making them better candidates for a so-called Pi sensing receptor. This group nicely showed that while isolated SPX domains can recognize 5-InsP_7_ and InsP_6_ with similar binding affinities, in the presence of their target transcription factor, plant SPX domains show a binding preference for PP-InsPs [65]. 

In plants, one SX domain-containing protein named SPX1 binds to PSR transcription factors, including PHR1 (Phosphate Starvation Response 1) and its homologs, which prevent the transcription factor from binding its target promoters under sufficient Pi conditions. Under low Pi conditions, the SPX1-PHR1 complex is unable to form, allowing the transcription factor to bind to its promoters and upregulate PSR genes [59,71] (Figure 2). PHR1 and PHL1 control the expression of a majority of the PSR genes, controlling numerous metabolic and development physiological adaptations to Pi deficiency. In rice, SPX proteins can associate with OsPHR2 in the presence of InsPs and PP-InsPs [65]. While initially Pi was proposed as the ligand for facilitating SPX-PHR complex formation, studies have identified InsPs, and more so PP-InsPs as bona fide ligands enabling this complex formation. The dissociation factor for Pi in the OsSPX4-OsPHR2 complex is over a factor greater than that of InsP_6_, while 5-InsP_7_ has the highest affinity of molecules tested, with a K_D_ in the low micromolar range [65,72]. Recently, two groups have characterized PPIPK double mutants (vih1/vih2) mutants, which as previously described, have significantly reduced levels of InsP_8_ and compromised Pi signaling pathways under Pi-sufficient levels [64,71]. In vivo analysis indicated that even though the intracellular Pi concentration was high in vih1/vih2 mutants, the SPX1-PHR1 interaction was compromised [62]. Additionally, in vitro binding analysis showed both InsP_7_ and InsP_8_ bind SPX1 in the micromolar range, with InsP_8_ exhibiting a higher binding affinity than InsP_7_, further indicating the importance of PP-InsP binding in mediating SPX-protein interactions [62].

Xpr1 (Xenotropic and polytropic retrovirus receptor 1) is the only SPX-domain containing protein identified in the human genome [52,65]. Xpr1 mediates Pi export in humans, and mutations within the SPX-domain of Xpr1 are responsible for primary familial brain calcification (PFBC) [59]. PFBC is characterized by calcification, predominantly of calcium phosphate, in the basal ganglia [59]. The mutations within the SPX-domain of Xpr1 cause a reduction in Pi export efficiency [73]. Li et al. has recently described the role of InsP_8_ in XPR1-mediated Pi efflux, providing a novel understanding of the genetic factors involved in defective bone maintenance and ectopic mineralization [73]. This study nicely showed how a reduction in InsP_8_ reduces Pi export by XPR1 and accelerates calcification and mineralization [73].

While the mechanism of InsP_8_-mediated XPR1 regulation in humans differs from the SPX-PHR1 interactions observed in plants, we can speculate that InsP_8_, and possibly PP-InsPs in general, may play common roles, facilitating protein-protein interactions or protein-PP-InsP interaction in various signaling pathways in animals and plants. It is important to note that *Arabidopsis* encodes 20 SPX-domain containing proteins, and rice encodes 15 predicted SPX-containing proteins, with a majority predicted to be involved in Pi homeostasis [59], although not all SPX proteins bind to transcription factors. The greatly expanded family of SPX genes present in plants in comparison to SPX genes in humans, and their involvement in Pi metabolism is likely due to the importance of Pi sensing and homeostasis in plant health. Given the important role Pi plays in cellular metabolism, understanding the regulation of Pi homeostasis via PP-InsPs is a critical aspect of how bioenergetic responses are mediated under various stimuli.

## 7. Concluding Remarks

Elucidating the biosynthesis, regulation, and function of PP-InsPs is crucial to furthering our understanding of energy metabolism and nutrient sensing in eukaryotes. In this review, we addressed four major questions remaining in the InsP and PP-InsP fields in both humans and plants; do animals have a Lipid-Independent InsP_6_ pathway, what is the catalytic flexibility of ITPKs, how is the biosynthesis of PP-InsPs regulated and what role PP-InsPs play in signaling pathways. ITPK1 has been recently characterized as a highly flexible enzyme, capable of catalyzing the phosphorylation of InsP_3_, InsP_4_, and InsP_6_ in *Arabidopsis*, and InsP_1_ through InsP_4_ in both humans and plants [30,40,41]. The recent advances in characterizing ITPK1, a key enzyme in the eukaryotic PP-InsP pathway, and the identification of ITPK1 homologs in Archaea, has provided insights into the evolutionary development and catalytic flexibility of InsP/PP-InsP kinases [30]. Further genetic and biochemical work on the multigene families that encode plant InsP kinases may continue to shed light on the evolution of InsP and PP-InsP pathways in other eukaryotes. The production of InsP_8_ via the unique bifunctional PPIP5Ks is a highly regulated process, with both the KD and PD acting interdependently to regulate InsP_7_ and InsP_8_ levels [52]. Further understanding of the activity and regulation of the PD, along with whether the IDR facilitates protein-protein interactions, will be crucial to understanding Pi sensing and homeostasis in plants and humans alike. The interaction between PP-InsPs and SPX-domain containing proteins is a phenomenon shared in humans and plants. While plants have an expanded group of SPX proteins, humans have only one SPX-domain containing protein identified to date [52,65]. Given the shared PP-InsP:SPX interactions observed in humans and plants, we speculate that PP-InsPs interact and facilitate protein-protein interactions with yet to be identified SPX proteins or structurally similar protein families. Identifying and characterizing these putative protein families could lead to novel insights into energy metabolism in humans. Given the known role of Ins and InsPs in numerous human diseases, it is important to consider how novel therapeutics and drugs targeting Ins and InsP biosynthetic pathways may impact PP-InsPs and their cellular functions.

## Figures and Tables

**Figure 1 molecules-25-02789-f001:**
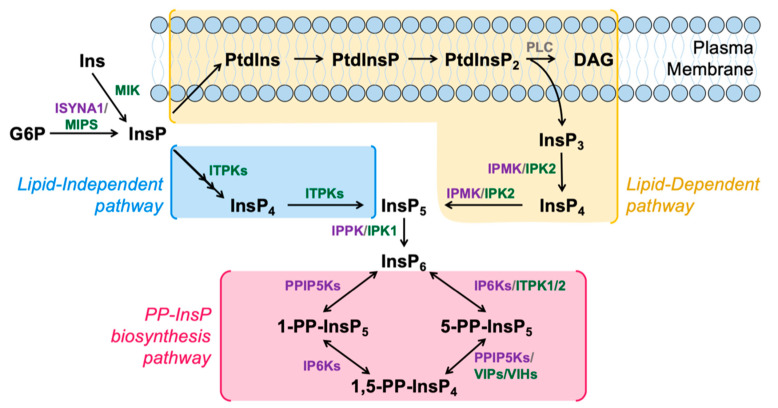
Schematic model of InsP and PP-InsP metabolism in humans and plants. InsPs are synthesized through the Lipid-Dependent (yellow) or Lipid-Independent (blue) pathways. PP-InsPs are synthesized from InsP_6_ via the PP-InsP biosynthesis pathway (pink). Key enzymes in the InsP and PP-InsP pathways conserved in animals and plants (grey), specific to animals (purple) or specific to plants (green) are shown next to their respective reactions. Enzyme names are defined in the text.

**Figure 2 molecules-25-02789-f002:**
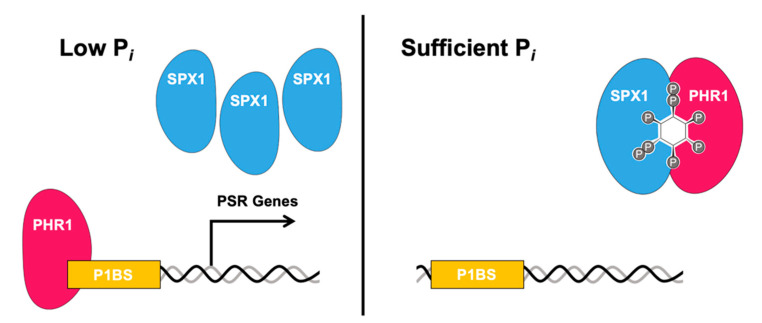
Model of PP-InsP regulation of the plant P*i* starvation response. Under low P*i* conditions (left), transcription factor PHR1 binds to P1BS-containing promoters. Under sufficient P*i* conditions, SPX1 interacts with PHR1 via PP-InsPs, preventing PHR1 from binding to P1BS-containing promoters. Adapted from [71].

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
