# Peer review of "Inositol Pyrophosphate Pathways and Mechanisms: What Can We Learn from Plants?"

_molecules, 2020, doi:10.3390/molecules25122789_

Round 1

Reviewer 1 Report

The current manuscript is nicely organized to review and highlight the recent progress regarding IP6 biosynthetic routes, control of PP-IP pathway, and Pi sensing via SPX domain proteins. The only minor suggestion is to add one key reference (Sci Signal. 2011 Aug 23;4(188):re1.) to the very beginning of introduction on the importance of PP-IPs in the field of human health.

Author Response

This reference has been added to the manuscript. Thank you for providing this.

Reviewer 2 Report

Inositol Pyrophosphate Pathways and Mechanisms: What can we learn from plants?

The answer is: A LOT.

The author wrote an elegant inositol phosphate review with an interesting angle comparing animal to plant. Some passages/ideas are questionable. This is not a criticism; it is far better reading a review where the authors are supporting a determinate view/s instead of a politically correct assay just listing all the literature available. The review is well written. Here I am listing a few issues that should be addressed.

1) Line 13: “InsP6, the most abundant InsP in plants.” IP6 is the most abundant IP in any organism analysed so far. I would delete -plant-.

2) Line 32: the concept that IP work using a ‘chemical signaling language [4]’ Is reminiscent of the inositol phosphates code originally expressed in (PMID: 16781889) that should be cited.

3) Line 38-39: In the sentence “… is the large phosphate (P) storage capacity of InsP6 present in plants.” I would add -seeds-.

4) Line 49: please change (4,5)P2  to  PstIns(4,5)P2.

5) Chapter 3. “Do animals have a lipid-independent InsP6 synthesis pathway?”  It is written as animals does not have the lipid-independent pathway but then in the next chapter, it is indicated that they do. It is confusing, I would change the chapter title to:

“Is the lipid-independent InsP6 synthetic pathway plant specific?” or similar sentence.

There are also some concepts that need to be rectified. While MIK are important to generate inositol-P from inositol in plant but not in animal, the conversion of glucose-6P to inositol-3P could generate the starting point of the lipid independent pathway in both plant and animal without the need of MIK. This important information is missing and should be added and the text (lines 82-84) should be reformulated. The concept expressed (line 95-97) that the plant enzyme responsible for IP2 phosphorylation is unknow is incorrect. The paper (PMID: 31754032) demonstrate that Arabidopsis and Oryza ITPK1 can convert IP to at least IP3-4 if not directly to IP5.

6) Figure 1: should indicate the conversion Glucose-6P to InsP and indicate ITPK1 as the enzyme responsible for the IP to IP3 conversion in the lipid-independent pathway. The experimental evidence that plant ITPKs are responsible for the conversion of IP to IP5 is as strong as the experimental evidence demonstrating that plant ITPKs work as an IP6K (an identical approach, rescues of yeast biochemical defect, was used to demonstrate both ITPKs activities). If in Fig.1 ITPKs are listed as IP6K it must also be listed as the enzyme responsible for the lipid-independent route.  The authors can’t use “two weights and two measures".

7) Line 105-107: CRISP knockout of HsIPPK decrease the level of IP6 but the level of IP5 does not change, it is IP4 that is going up (PMID: 31851928). There is plenty of unknow in plant and mammal IP pathway.  

8) Line 141: “canonical InsP3 kinases (IP3Ks)” should be referenced as “canonical InsP3 3-kinase (IP3 3K)” (PMID:20066467)

9) Line 144: “accommodate InsP3, InsP4, InsP6 substrates” change to “accommodate several diverse substrates” or list InsP to InsP6.

10) Line 182: Here reference [46] is inappropriate. It is better to cite a recent paper that has actually measured IP6-7-8 in Dictyostelium (PMID:24416420)

11) Line 180-190 Section PP-IP5K Phosphatase Domain. This section should give credit to Fleig’s laboratory that was the first to characterize the phosphatase domain of Asp1 (PMID:25254656; PMID:29440310). Credit should also be given to recent York’s lab paper (PMID:32303658).

12) Line 268: please delete ‘To date’

13) Chapter 7. Concluding Remarks should be slightly modified to be coherent with the above suggestion. I would conclude by answering the title! Plant could teach us a lot (or similar sentences).

Author Response

We appreciate the thoughtful and informative comments. We recognize that the Reviewer is clearly knowledgeable and passionate about this pathway, which helped to make our review stronger.

Here is how we addressed this Reviewer's comments (our responses in italics):

1) Line 13: “InsP6, the most abundant InsP in plants.” IP6 is the most abundant IP in any organism analysed so far. I would delete -plant-.

We deleted the word last....now on line 13.

2) Line 32: the concept that IP work using a ‘chemical signaling language [4]’ Is reminiscent of the inositol phosphates code originally expressed in (PMID: 16781889) that should be cited.

We now give appropriate credit to John York on new line 32 by citing the suggested reference, and then referring to the plant review thereafter. I (Gillaspy) am embarrassed and realize now that I unintentionally co-opted John's language!

3) Line 38-39: In the sentence “… is the large phosphate (P) storage capacity of InsP6 present in plants.” I would add -seeds-.

This change was made and is on line 40.

4) Line 49: please change (4,5)P2  to  PstIns(4,5)P2.

Change made; on line 51.

5) Chapter 3. “Do animals have a lipid-independent InsP6 synthesis pathway?”  It is written as animals does not have the lipid-independent pathway but then in the next chapter, it is indicated that they do. It is confusing, I would change the chapter title to:

“Is the lipid-independent InsP6 synthetic pathway plant specific?” or similar sentence.

This change was made....line 68.

There are also some concepts that need to be rectified. While MIK are important to generate inositol-P from inositol in plant but not in animal, the conversion of glucose-6P to inositol-3P could generate the starting point of the lipid independent pathway in both plant and animal without the need of MIK. This important information is missing and should be added and the text (lines 82-84) should be reformulated. The concept expressed (line 95-97) that the plant enzyme responsible for IP2 phosphorylation is unknow is incorrect. The paper (PMID: 31754032) demonstrate that Arabidopsis and Oryza ITPK1 can convert IP to at least IP3-4 if not directly to IP5.

6) Figure 1: should indicate the conversion Glucose-6P to InsP and indicate ITPK1 as the enzyme responsible for the IP to IP3 conversion in the lipid-independent pathway. The experimental evidence that plant ITPKs are responsible for the conversion of IP to IP5 is as strong as the experimental evidence demonstrating that plant ITPKs work as an IP6K (an identical approach, rescues of yeast biochemical defect, was used to demonstrate both ITPKs activities). If in Fig.1 ITPKs are listed as IP6K it must also be listed as the enzyme responsible for the lipid-independent route.  The authors can’t use “two weights and two measures".

Lines 96-110 address Reviewer points 5 and 6: We rectified the identified concepts centered around our description of the MIK and the unknown InsP2 kinase, along with that of the ITPK’s newfound catalytic activities.  We admit that we overgeneralized as to the importance of MIK, and so we added Reviewer 2’s point about MIPS. We do want to point out that although we overgeneralized from plants, the fact that mik mutants have reduced InsP6 in seed does show that MIK is required for the lipid-independent pathway in plant seeds.  But we admit, as the review noted, that this doesn’t mean that humans require an MIK for a lipid-independent pathway. We also corrected our thought that the plant enzyme responsible for InsP2 phosphorylation is unknown, and tried to “use a single weight and measure” in describing the pathway, given the new data on the ITPKs.  We also correspondingly altered Figure 1 to reflect this change.

7) Line 105-107: CRISP knockout of HsIPPK decrease the level of IP6 but the level of IP5 does not change, it is IP4 that is going up (PMID: 31851928). There is plenty of unknow in plant and mammal IP pathway.  

Lines 116-117 contain our changes...." The human IPK1, named HsIPPK, can be knocked down, and these mutants accumulate InsP4 at the expense of InsP6 levels [35], which is a notable difference from the situation in Atipk1 mutants." 

8) Line 141: “canonical InsP3 kinases (IP3Ks)” should be referenced as “canonical InsP3 3-kinase (IP3 3K)” (PMID:20066467)

We made the suggested change and added the suggested reference. We left the abbreviation as IP3K as this is what is used in other published papers.

9) Line 144: “accommodate InsP3, InsP4, InsP6 substrates” change to “accommodate several diverse substrates” or list InsP to InsP6.

Line 143: We went with “accommodate several diverse substrates”

10) Line 182: Here reference [46] is inappropriate. It is better to cite a recent paper that has actually measured IP6-7-8 in Dictyostelium (PMID:24416420) Replace reference

Line 179: we removed the inappropriate reference and added the suggested references.

11) Line 180-190 Section PP-IP5K Phosphatase Domain. This section should give credit to Fleig’s laboratory that was the first to characterize the phosphatase domain of Asp1 (PMID:25254656; PMID:29440310). Credit should also be given to recent York’s lab paper (PMID:32303658).

New line 183; added the suggested references.  

12) Line 268: please delete ‘To date’

Line 267: Deleted this.

13) Chapter 7. Concluding Remarks should be slightly modified to be coherent with the above suggestion. I would conclude by answering the title! Plant could teach us a lot (or similar sentences).

Lines 285-309: We added the suggested phrase and then highlighted ITPK1 as one example. The sentence on identifying unknown enzymes was modified....Line 294: Further genetic and biochemical work on the multigene families that encode plant InsP kinases may shed light on the evolution of InsP and PP-InsP pathways in other eukaryotes.

Reviewer 3 Report

It's always nice to review a well-written paper and this one is a good example of that! It provides a very complete overview of current knowledges on the InsPP biosynthesis and their action mode. The authors are well versed in their subject matter and I have only very minor comments.

On figure 1 for InsP8 production the right part indicates plant/animal enzymes but not left one (for example VIPs/VIHs can promotes 1-PP-insP5 production). I also do not see the production at the beginning of the chain of InsP1 produced through the activity of ISYNA1 from glucose. Authors mentioned that ITPK1 can phosphorylate InsP until InsP3 but in the reference used (Desfougères et al.,2019, PNAS) it can even reach InsP5.

Authors indicated (L250) that in Plants SPX bind to phosphate starvation response transcription factors. This is fully exact but there is also many example of additional activities of SPX domain containing proteins not linked with TF such as control of leaf inclination, phosphate transport (PHO1) or protein degradation (NLA)…

L261 Analysis of vih1/vih2 double mutant is linked with reference 55 but forget the reference 54. This article was published the same year and also produced a great genetic work on this aspect.

L280 InsP8 is linked with signaling pathways but it also control Pi flux (through PHO1 at least) so is ion flux a signaling pathways? It is a puzzling question when this ion itself is at the origin of a signaling cascade. Should it be highlighted or the authors consider that the signal path includes this aspect? I leave it to them to judge.

Author Response

We thank the reviewer for the compliment, and for your time on this review. 

Our responses are in italics:

On figure 1 for InsP8 production the right part indicates plant/animal enzymes but not left one (for example VIPs/VIHs can promotes 1-PP-insP5 production). I also do not see the production at the beginning of the chain of InsP1 produced through the activity of ISYNA1 from glucose. Authors mentioned that ITPK1 can phosphorylate InsP until InsP3 but in the reference used (Desfougères et al.,2019, PNAS) it can even reach InsP5.

We did not include VIPs/VIH enzymes acting on InsP6 in Figure 1 because that is not the major pathway in non-plant organisms, and in plants, our biochemical data did not support InsP6 as a major substrate either. We amended Figure 1 to include ISYNA1 and the ITPKs. For the point on ITPK1, we changed the text which is now found at Line 107-108 and includes the point raised on production of InsP5.

Authors indicated (L250) that in Plants SPX bind to phosphate starvation response transcription factors. This is fully exact but there is also many example of additional activities of SPX domain containing proteins not linked with TF such as control of leaf inclination, phosphate transport (PHO1) or protein degradation (NLA)…

Yes, this is a great point. In Line 277-278: we made note of this important point that…..” It is important to note that Arabidopsis encodes 20 SPX-domain containing proteins, and rice encodes 15 predicted SPX-containing proteins, with a majority predicted to be involved in Pi homeostasis [64], although not all SPX proteins bind to transcription factors.”

L261 Analysis of vih1/vih2 double mutant is linked with reference 55 but forget the reference 54. This article was published the same year and also produced a great genetic work on this aspect.

Line 259: We apologize for this mistake, and have corrected the referencing on this point.

L280 InsP8 is linked with signaling pathways but it also control Pi flux (through PHO1 at least) so is ion flux a signaling pathways? It is a puzzling question when this ion itself is at the origin of a signaling cascade. Should it be highlighted or the authors consider that the signal path includes this aspect? I leave it to them to judge.

This is such a great question........perhaps the answer is that an environmental signal (Pi) stimulates production of intracellular message (InsP8), that then signals to the SPX domain on PHO1 that ultimately impacts aspects of Pi uptake. I would say -Yes- this is signaling! I have made too many speculations in describing my opinion, and I suspect all three reviewers have indulged us perhaps to their limit, in terms of speculating in this review. I think this could be the focus of another review on transport, and it is really a fascinating question.